# The BCG Moreau Vaccine Upregulates In Vitro the Expression of TLR4, B7-1, Dectin-1 and EP2 on Human Monocytes

**DOI:** 10.3390/vaccines11010086

**Published:** 2022-12-30

**Authors:** Paulo R. Z. Antas, Andreon S. M. da Silva, Lawrence H. P. Albuquerque, Matheus R. Almeida, Evelyn N. G. S. Pereira, Luiz R. R. Castello-Branco, Carlos G. G. de Ponte

**Affiliations:** 1Laboratório de Imunologia Clínica, Instituto Oswaldo Cruz, Fiocruz, and Instituto Nacional de Ciência e Tecnologia em Tuberculose (INCT-TB), Rio de Janeiro 21040-900, Brazil; 2Department of Infectious Diseases, Leiden University Medical Centre, 2333ZA Leiden, The Netherlands; 3Laboratório de Investigação Cardiovascular, Instituto Oswaldo Cruz, Fiocruz, Rio de Janeiro 21040-900, Brazil

**Keywords:** BCG vaccine, tuberculosis, monocyte, innate immunity, biomarker

## Abstract

Background: Tuberculosis (TB) is currently the second greatest killer worldwide and is caused by a single infectious agent. Since Bacillus Calmette–Guérin (BCG) is the only vaccine currently in use against TB, studies addressing the protective role of BCG in the context of inducible surface biomarkers are urgently required for TB control. Methods: In this study, groups of HIV-negative adult healthy donors (HD; *n* = 22) and neonate samples (UCB; *n* = 48) were voluntarily enrolled. The BCG Moreau strain was used for the in vitro mononuclear cell infections. Subsequently, phenotyping tools were used for surface biomarker detection. Monocytes were assayed for TLR4, B7-1, Dectin-1, EP2, and TIM-3 expression levels. Results: At 48 h, the BCG Moreau induced the highest TLR4, B7-1, and Dectin-1 levels in the HD group only (*p*-value < 0.05). TIM-3 expression failed to be modulated after BCG infection. At 72 h, BCG Moreau equally induced the highest EP2 levels in the HD group (*p*-value < 0.005), and higher levels were also found in HD when compared with the UCB group (*p*-value < 0.05). Conclusions: This study uncovers critical roles for biomarkers after the instruction of host monocyte activation patterns. Understanding the regulation of human innate immune responses is critical for vaccine development and for treating infectious diseases.

## 1. Background

Tuberculosis (TB) is one of the ten leading causes of death worldwide, with an estimated 1.7 billion people infected [1]. The TB etiologic agent *Mycobacterium tuberculosis* (Mtb) causes disease in about 5 and 10% of these Mtb-infected individuals. In 2020, about 10 million people contracted TB worldwide, and 1.3 million infected people died, roughly 100,000 more than in the previous year [1]. To control this disease, studies on TB vaccines are urgently needed.

The Bacillus Calmette–Guérin (BCG) vaccine was developed a century ago to tackle TB [2], and immunisation is one of the factors that primarily reduces the risk of TB death among children and young adults [1,3]. In countries where BCG vaccination is part of EPI nationwide campaigns, this vaccine reaches a milestone of preventing about 40,000 TB deaths yearly [4]. In addition, the BCG vaccine may be seen as inducing an agnostic or heterologous immunity [5,6] during sepsis [7], meningitis [8], influenza [9], leukaemia [10], asthma [11], allergies [12], type-2 diabetes [13], in bladder cancer treatment [14], and more recently, COVID-19 [15]. While different BCG strains with different immunogenicity and residual virulence have grown worldwide [16,17], the BCG Moreau strain arrived in in 1927 and has been used in Brazil ever since [18,19].

Several Toll-like Receptors (TLRs), such as TLR4 (CD284), and C-type Lectin Receptors (CLRs), such as Dectin-1 (DEC-1 or CD369), are involved in the first recognition of Mtb antigens and during the activation of immune cells [20,21,22,23,24]. TLR4 has been described as induced on the surface of host phagocytes after infection with virulent Mtb strains, such as H37Rv [25,26], and shedding in its soluble form by ADAM17 cleavage in pleural effusion from TB patients [27]. On the other hand, seminal studies have shown that TLR2-mediated production of TNF requires DEC-1 expression during the activation of macrophages in mycobacterial infection [28,29,30]. In macrophages of adults and neonates, the simultaneous activation of DEC-1 and TLRs by different agonists was able to induce the production of IL-12p70 in dendritic cells, leading to the polarisation of Th1 cells and the expression of costimulatory molecules, such as CD40, B7-1 (CD80), B7-2 (CD86), and HLA-DR [31,32]. Strikingly, another study showed that TLR4 agonists increased the ability of antigen-presenting cells (APCs) to present in vitro BCG vaccine antigens to T cells, which correlated with increased HLA-DR expression [33].

The B7 family is one of the best-characterised groups of molecules playing a role in the co-stimulation and homeostasis of immune responses, being an interface between innate and adaptive immunity, leading to either activation or tolerance [34,35,36,37]. The activation of Th1 cells is critical for TB control [38,39], and it has been observed that blocking B7-1 during *M. bovis* infection reduces IFN-γ secretion from T lymphocytes [40].

Prostaglandin E2 (PGE2) is a bioactive eicosanoid molecule derived from arachidonic acid produced by APCs and is involved in distinct immune responses [41,42,43]. Studies have shown that PGE2 levels increase in vitro during both Mtb [43] and BCG Moreau vaccine infections [44]. Among several other functions [45,46], the combination of PGE2, IL-1β, and IL-23 promotes IL-17 production [47]. In professional APCs, only EP2 is activated for cytokine production [45], being the most abundant of the four PGE2 receptors [48]. Of note, PGE2 levels regulate EP2 expression [49].

The T cell immunoglobulin and mucin-domain-3 (TIM-3 or CD366) is a surface molecule expressed by Th1, Tc1, Th17, and Treg cells [50,51,52]. It has been shown in bidirectional contexts, sometimes inhibitory, but it can also promote immune activation to produce essential cytokines [53,54]. However, studies of TIM-3 expression in mycobacterial diseases are lacking.

The immune responses after BCG vaccination differ between healthy adults and neonates [55,56]. The immunomodulatory capacity of the BCG Moreau vaccine to induce some of the well-characterised innate immunity molecules on monocytes from those individuals, such as TLR4, B7-1, DEC-1, EP2, and TIM-3, has not yet been studied. Thus, to address the remaining critical issues beyond these premises, it is essential to compare the in vitro immune response profiles induced by the BCG vaccine in individuals, both already BCG-sensitised and naïve leukocytes, to help understand how innate immunity induces long-lasting protection after stimuli.

## 2. Material and Methods

### 2.1. Study Participants

Two groups of donors were enrolled for this study: healthy adults (HD; *n* = 22) at a public blood bank (anonymous donations from individuals aged ≥ 18 years) and healthy mothers (aged ≥ 18 years) who participated in procedures to puncture umbilical cords and obtain blood samples from newborns’ umbilical cord blood (UCB; *n* = 48). Since 1967, Brazil has had a policy of providing universal BCG vaccination soon after birth. Strict definitions of those cohorts and inclusion and exclusion criteria for those HIV-seronegative individuals are described elsewhere [57]. However, we acknowledge that maternal BCG scar rather than just a history of receipt of BCG is likely to modify an infant’s response to BCG [58]. Hence, due to the importance of BCG and other mycobacteria exposure in a given population, we were unable to determine the BCG status of our HD cohort (or even to perform additional tests) due to the blood bank depository guidelines.

### 2.2. Mononuclear Cells Purification, Culture, In Vitro Infection with BCG and Phenotyping

The peripheral blood mononuclear cells (PBMC) and cord blood mononuclear cells (CBMC) were separated within no more than 24 h (average of 5 h) of obtaining blood specimens from all the study participants and cultured as previously described [59]. Briefly, cultures of freshly isolated PBMCs and CBMCs (1 × 10^6^ cells each) in RPMI medium (Sigma Immunochemicals, St. Louis, MD, USA) were supplemented with 10% foetal bovine serum (FBS) only (baseline), and those in vitro infected with the BCG Moreau vaccine (at a single dose, individual batches of sealed glass vials containing liquid suspension with approximately 1 × 10^7^ viable bacilli), performed at a multiplicity of infection (MOI) of 2:1 (bacilli/host cell), were stored at 37 °C in a humidified 5% CO_2_ atmosphere in individual 12 × 75 mm sterile polystyrene tubes (Falcon, Corning Inc., Corning, NY, USA) initially for 24 h, 48 h, or 72 h. Notably, the respective master batches of the BCG vaccine were used once for each infection and then discarded. After cells were washed in a buffer solution (PBS with 0.01% sodium azide-NaN3 and 0.1% bovine serum albumin (BSA; Sigma-Aldrich)), 1% goat serum was added (CECAL/FIOCRUZ) to avoid unspecific binding, followed by incubation for 10 min at room temperature (RT). Afterwards, the cells suspended in 50 μL of the same buffer were subjected to labelling with anti-human CD14 conjugated to PE.CY7 (Biolegend, San Diego, CA, USA), anti-human CD284 (TLR4) conjugated to PE (Biolegend), anti-human CD80 (B7-1) conjugated to FITC (Thermo Scientific^®^, Waltham, MA, USA), anti-human CD369 (CLec7a/DEC-1) conjugated to FITC (Thermo Scientific, Waltham, MA, USA), anti-human EP2 conjugated to PE (Biolegend), and anti-human CD366 (TIM-3) conjugated to BV421 (Biolegend), all incubated for 30 min at 4 °C in the dark. Finally, the cells were washed twice in the same buffer and fixed with PBS plus 2% paraformaldehyde. The cells were then analysed using CytoFLEX S (Beckman Coulter, Brea, CA, USA) or FACSAria II (BD Biosciences, Franklin Lakes, NJ, USA) flow cytometer devices. About 10,000 events were acquired in the region of interest using CytExpert (Beckman Coulter) or CellQuest softwares (BD Biosciences). Simple labelling for fluorochrome compensation was carried out using commercial calibration microspheres (OneComp eBeads, eBiocience, San Diego, CA, USA). FlowJo 10 software (TreeStar, Woodburn, OR, USA) was used to collect and analyse flow cytometry data. The gate strategy used is depicted to avoid fluorescence overlap between the panels (Appendix A). The percentage of positive cells for each marker was recorded, as well as the per-cell marker expression utilising the median fluorescence intensity (MFI) of the cell population, which was normalised to LOG10 to rule out voltage differences between devices after different sample readings.

### 2.3. Statistical Evaluation

Statistical analyses were performed using Prism software, version 7 (GraphPad Software, San Diego, CA, USA). For the kinetic studies, each BCG-stimulated condition was subtracted from the non-infected control (baseline), generating the delta. Data transformation was used for the MFI data before a given statistical test and tested for normal distribution. The normality of the distribution of variables was evaluated with Shapiro–Wilk tests and confirmed with Q–Q graphs. The Mann–Whitney U test was used to compare the baseline versus BCG-infected cells. The Kruskal–Wallis test was used to compare the variations during the kinetic expression. *p* < 0.05 was considered statistically significant.

## 3. Results

### 3.1. The BCG Moreau Vaccine Modulates In Vitro the Expression of TLR4, B7-1, DEC-1, and EP2 on Human Monocytes from Healthy Adult Donors

To systematically evaluate robust biomarkers for human macrophage subsets in vitro, flow cytometric immunophenotyping is considered an indispensable tool [60]. In order to ascribe the best timeline related to each biomarker assessed, we first analysed the in vitro kinetics of TLR4, B7-1, DEC-1, TIM-3, and EP2 expressions on monocytes induced by the BCG Moreau vaccine at 24 h, 48 h, and 72 h from five HD individuals. Accordingly, the BCG Moreau induced an increase in cells expressing all markers evaluated, although not statistically significant (Figure 1). In addition, there was a parallel increase in the absolute number of cells expressing these biomarkers. The TLR4, B7-1, DEC-1, and TIM-3 expressions increased at 48 h, except for EP2, which showed a marked increase at 72 h (Appendix A). Hence, these two timelines were chosen to perform the subsequent phases of this study.

To expand the prior observed data, we enlarged the HD group (a total of 17 donors) and included the neonate (UCB) group in parallel (a total of 48 donors). After infection, the flow cytometric analysis revealed that the BCG Moreau vaccine was indeed able to induce a higher (*p* < 0.05) frequency of cells expressing TLR4 (almost twice), B7-1 (almost three-fold), and more strikingly, both DEC-1 and EP2 (almost four-fold each) on monocytes from the HD group (Figure 1A; Appendix A). The frequency of monocytes expressing TIM-3 after BCG infection did not achieve statistical significance. Finally, we failed to show any ability of the BCG Moreau vaccine to modulate the expression of these same phenotypic markers per cell (Figure 1B; Appendix A). We also failed to show the modulatory effect of this vaccine in the UCB group.

### 3.2. The BCG Moreau Vaccine Induces a Differential Expression of TLR4, B7-1, and DEC-1 on Human Monocytes from Neonates vs. Adults

After assessing the in vitro expression of phenotypic markers using distinct cohorts of healthy neonates and adults, the constitutive (baseline) and induced levels (BCG-infected) were compared. The constitutive B7-1 and DEC-1 levels showed a five-fold increase (*p* ≤ 0.001) and almost a four-fold increase (*p* = 0.01), respectively, when UCB versus HD groups were compared (Figure 2; Appendix A). However, HD individuals showed a constitutively higher intensity for both TLR4 (*p* = 0.01) and B7-1 expressions (*p* = 0.01; Figure 2; Appendix A). Significantly, the frequency of monocytes expressing EP2 after in vitro infection with the BCG Moreau vaccine was five-fold higher in the HD than in the UCB groups (*p* < 0.001; Figure 2; Appendix A).

## 4. Discussion

Neonates rely on innate immunity until their immune system fully develops [61]. Although Yan and colleagues [62] reported that adults and neonates have similar in vitro expression of TLR4 upon baseline conditions—a piece of data corroborated in our current study—we have also shown that adults constitutively expressed the higher intensity of TLR4 per cell than neonates. In addition, differences in TLR4 function and regulation in the UCB group, such as a lower sensitivity to stimuli via TLRs that lead to reduced TNF production, were described later [63,64,65,66]. Hence, it must be verified later whether those differences are related to the lowest baseline TLR4 expression per cell in the UCB group. In this study, although we found that the BCG Moreau vaccine could induce strong TLR4 expression in BCG-sensitised monocytes in vitro, we failed to show any similar modulatory effect on cells from the UCB group. In cases of severe infection that leads to sepsis, TLR4 is upregulated [67], but another study reported a weak expression of TLR4 in low birth weight neonates [68]. These authors argue that this may indicate greater susceptibility to Gram-negative bacterial infections due to a lack of inflammatory cytokines to enhance the incipient immune response.

Here, we hypothesised that increased expressions of TLR4 and DEC-1 may demonstrate one of the critical pathways used by the BCG Moreau vaccine to provide faster and more effective recognition of pathogens by BCG-sensitised adults. Both molecules ultimately lead to both TNF and IFN-γ secretion, and an in vitro induction of TLR4 by the BCG vaccine has been found on human macrophages as well, but in contrast to our findings, it peaked at 72 h [69,70]. Although Lichte and colleagues [71] stated that preliminary changes in TLR4 expression on circulating monocytes are not necessarily part of an inflammatory response, additional studies show this induction lasts for at least one year [72]. This fact strongly supports epigenetic reprogramming in the innate immune cell compartment [73].

It has been broadly accepted that different BCG vaccines induce different immune responses in humans. However, our data on in vitro increased B7-1 levels found in BCG Moreau-sensitised adults corroborate another study using the BCG Japan vaccine in humans [74]. It has been hypothesised that the BCG substrains from group I (Moreau, Russia, and Japan) hold high immunogenicity, although there are virtually no surrogate markers to predict vaccine efficacy [75]. Moreover, the induction of B7-1 levels may lead to a more efficient antigen presentation [35], which not only might be beneficial in the context of TB but mainly supports an elegant study of BCG vaccine and dengue virus co-stimulation, a clear agonistic, heterologous effect [76].

Our prior study has already shown that the BCG Moreau vaccine induced in vitro PGE2 expression in both the HD and UCB groups after 48 h of mononuclear infection [44]. Therefore, we initially postulated that a concomitant modulation in EP2 levels might follow that soon after, and this was conclusively confirmed here at 72 h. Similarly to the increase in BCG-induction B7-1 levels, this high EP2 expression might demonstrate a better antigenic presentation by a cross-priming mechanism, a differential advantage during vaccination [77].

In this study, we failed to find any significant in vitro induction by the BCG vaccine of the five phenotypic markers in the neonate UCB group. However, the higher constitutive values of B7-1 and DEC-1 suggest that those in vitro levels were already increased at birth. This increase may be related to a remnant of maternal immunity [78,79], which may have masked the induction by the BCG Moreau vaccine. Additionally, during the comparison of the HD versus UCB groups, the analysis of constitutively expressed levels revealed that, compared to adults, neonates had five-fold cells expressing B7-1 and almost four-fold cells expressing DEC-1. In addition, we found that adults had a superior number of TLR4 and B7-1 molecules in their monocytes. If there is a compensation mechanism between the number of positive cells (%) and the intensity of expression in the number of markers per cell (MFI), it remains to be determined. Most in vitro studies on the BCG vaccine have ignored this distinction. Still, in this comparison, we also observed that the BCG Moreau vaccine induced more EP2-positive cells in adults. This induction of EP2 is explained by the greater ability of adult cells to respond to inflammatory stimuli [80,81]. Thus, the induction of EP2 could be more related to the maturity of the immune system than to the co-expression of prostaglandin, since the expression of the mediator did not also mean an increase in the EP2 receptor in a mouse model [82].

A first limitation of the present study was the lack of demographical data. In the HD group, adult samples were obtained from individuals from whom there was no available information on tuberculin skin test positivity. As stated earlier, the anonymous donations policy was enforced by the IRB approvals precluding records of clinical characteristics. Still, the only secure information is that Brazilian individuals have been vaccinated with the Moreau BCG strain only. Second, using the strain specificity topic, it was necessary to test our hypothesis using an alternative cohort of matched individuals already vaccinated with any non-related BCG Moreau vaccine elsewhere. Third, although we were unable to explore further in this study, we benefited from the use of pre-exposure markers to mycobacteria, thus allowing us to relate the stimulus provided by the BCG vaccine to any eventual recall immune response to mycobacteria [83]. Additional studies with increased sample sizes are warranted to confirm the present results.

Taken together, our study presented several examples of surface biomarkers in mononuclears that have been broadly modulated in vitro by the BCG vaccine, data consistent with previous findings. Overall, these results reinforce that the BCG Moreau vaccine is a potent immunostimulating agent. Thus, this study uncovers critical roles for biomarkers after the instruction of host monocyte activation patterns.

## 5. Conclusions

The BCG Moreau vaccine induced in vitro a significant increase in the expression of all the phenotypic markers studied, except TIM-3, in monocyte populations of healthy adults. When comparing adults and neonates, the induced in vitro expression of these markers was shown to be differential only for EP2. However, a differential character of constitutive expression was observed for TLR4, B7-1, and DEC-1. The increase in these markers in the acute infection model demonstrated that the BCG Moreau vaccine activates the immune response mediated by important surface receptors on the monocytes of adult individuals, and such activation may contribute to the generation of a protective cellular immune response. Understanding the regulation of human innate immune responses is critical for vaccine development and for treating infectious diseases.

## Figures and Tables

**Figure 1 vaccines-11-00086-f001:**
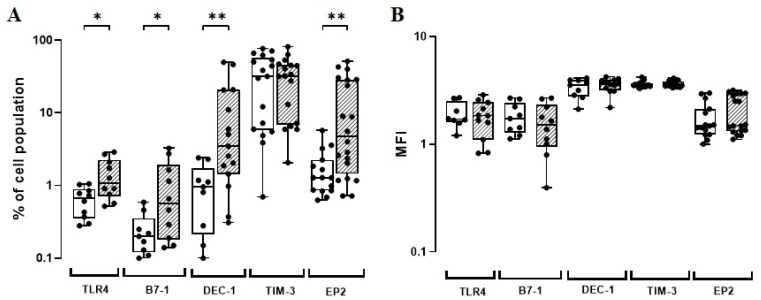
(**A**) TLR4, B7-1, DEC-1, TIM-3, and EP2-phenotypic marker expression levels (%) induced in vitro by BCG Moreau (hatched) and baseline (opened) at 72 h (EP2) or 48 h (others) on human mononuclears from adult healthy donors (HD; *n* = 17). Box and whisker plots denote the median and interquartile range phenotypic marker expression values in each condition. * *p* ≤ 0.05; ** *p* < 0.01. (**B**) The same as (**A**), except for the phenotypic marker intensity (MFI) induced under similar conditions.

**Figure 2 vaccines-11-00086-f002:**
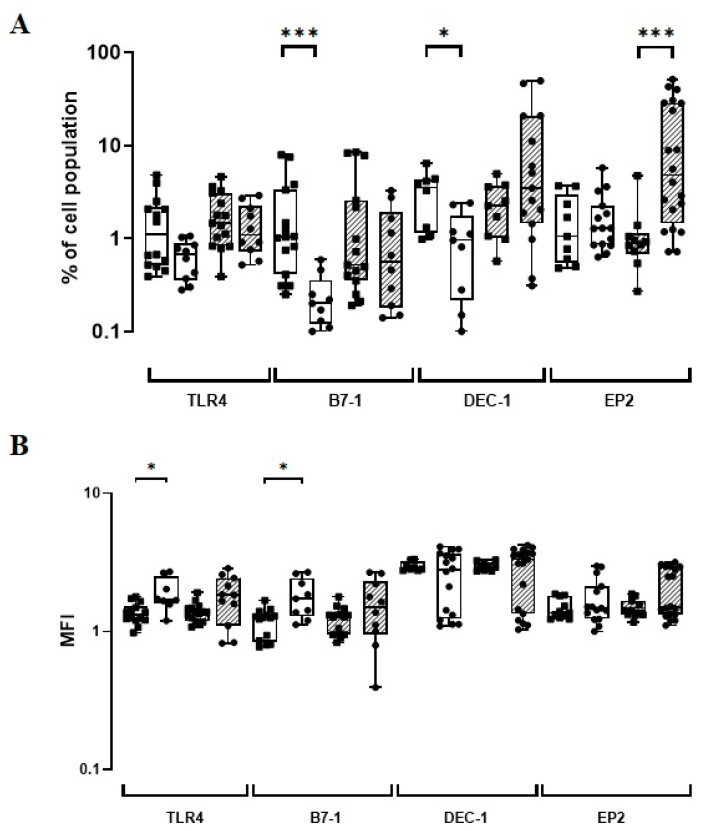
(**A**) The TLR4, B7-1, DEC-1, and EP2-phenotypic marker expression levels (%) induced in vitro by BCG Moreau (hatched) and baseline (opened) at 72 h (EP2) or 48 h (others) on human mononuclears from healthy adult (dot) and neonate (square) individuals. Box and whisker plots denote the median and interquartile range phenotypic marker expression values in each condition. (**B**) The same as for (**A**), except for the phenotypic marker intensity (MFI) induced under similar conditions. * *p* ≤ 0.05; *** *p* < 0.001.

## Data Availability

The data used to support the findings of this study are available from the corresponding author upon request.

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
