# Peer review of "The BCG Moreau Vaccine Upregulates In Vitro the Expression of TLR4, B7-1, Dectin-1 and EP2 on Human Monocytes"

_vaccines, 2022, doi:10.3390/vaccines11010086_

Round 1
Reviewer 1 Report
The manuscript by Antas et al. “The BCG Moreau vaccine upregulates in vitro the expression of 2TLR4, B7-1, Dectin-1 and EP2 on human monocytes” describes the evaluation of BCG vaccination in to induce monocyte activation using biomarkers as TLR4, B7-1, Dectin-1, EP2, and TIM-3. The results are quite interesting and show that some biomarkers were upregulated in monocytes humans compared with neonate samples and for this are suitable for publication. I have some minor suggestions to improve the manuscript.BACKGROUND
Line 36: in the sentence “To control this plague” please change plague for disease.
RESULTS:
I suggest improving figure 2, it is difficult to differentiate circles (human mononuclear cells from healthy adults) from squares and (neonates). My suggestion is to fill circles or squares in the graph with color (or in black). The unfilled shapes make it quite hard to visualition the data.
Author Response
A. We appreciate the referee's suggestion to improve the figures definition. Please, see the attachment and find attached the 2 figures in better resolutions.

Reviewer 2 Report
Dear Authors
I was very pleased to review this manuscript.
This study proved to be important as it adds to the knowledge of understanding the regulation of human innate immune responses, which will help in the development of vaccines and treatment of infectious diseases.
It is a subject that becomes relevant of today.
It adds a few more findings when compared to other studies.
The groups under study were well chosen.
As for the results, they would benefit if they were also presented in a table. I would suggest that at least two tables summarizing the results (Mann-Whitney and Kruskal-Wallis tests) be prepared.
The discussion and conclusions are sufficiently developed.
The references are adequate.
My Best Regards
Author Response
A: We appreciate the referee's suggestion to better present our data. Please, find attached the novel, 2 tables with data compiled as supplementary files.
